# Dock10 Regulates Cardiac Function under Neurohormonal Stress

**DOI:** 10.3390/ijms23179616

**Published:** 2022-08-25

**Authors:** Liad Segal, Sharon Etzion, Sigal Elyagon, Moran Shahar, Hadar Klapper-Goldstein, Aviva Levitas, Michael S. Kapiloff, Ruti Parvari, Yoram Etzion

**Affiliations:** 1Department of Physiology and Cell Biology, Ben-Gurion University of the Negev, Beer-Sheva 8410501, Israel; 2Regenerative Medicine and Stem Cell (RMSC) Research Center, Ben-Gurion University of the Negev, Beer-Sheva 8410501, Israel; 3Pediatric Cardiology Department, Soroka University Medical Center, Beer-Sheva 8410501, Israel; 4Departments of Ophthalmology and Medicine (Cardiovascular Medicine), Stanford Cardiovascular Institute, Stanford University, Palo Alto 94304-1209, CA, USA; 5The Shraga Segal Department of Microbiology, Immunology and Genetics, Ben-Gurion University of the Negev, Beer-Sheva 8410501, Israel; 6National Institute for Biotechnology in the Negev, Ben-Gurion University of the Negev, Beer-Sheva 8410501, Israel

**Keywords:** Rho GTPases, Cdc42, Rac1, MAPK signaling, pathological cardiac hypertrophy

## Abstract

Dedicator of cytokinesis 10 (Dock10) is a guanine nucleotide exchange factor for Cdc42 and Rac1 that regulates the JNK (c-Jun N-terminal kinase) and p38 MAPK (mitogen-activated protein kinase) signaling cascades. In this study, we characterized the roles of Dock10 in the myocardium. In vitro: we ablated Dock10 in neonatal mouse floxed *Dock10* cardiomyocytes (NMCMs) and cardiofibroblasts (NMCFs) by transduction with an adenovirus expressing Cre-recombinase. In vivo, we studied mice in which the *Dock10* gene was constitutively and globally deleted (Dock10 KO) and mice with cardiac myocyte-specific *Dock10* KO (Dock10 CKO) at baseline and in response to two weeks of Angiotensin II (Ang II) infusion. In vitro, Dock10 ablation differentially inhibited the α-adrenergic stimulation of p38 and JNK in NMCM and NMCF, respectively. In vivo, the stimulation of both signaling pathways was markedly attenuated in the heart. The Dock10 KO mice had normal body weight and cardiac size. However, echocardiography revealed mildly reduced systolic function, and IonOptix recordings demonstrated reduced contractility and elevated diastolic calcium levels in isolated cardiomyocytes. Remarkably, Dock10 KO, but not Dock10 CKO, exaggerated the pathological response to Ang II infusion. These data suggest that Dock10 regulates cardiac stress-related signaling. Although Dock10 can regulate MAPK signaling in both cardiomyocytes and cardiofibroblasts, the inhibition of pathological cardiac remodeling is not apparently due to the Dock10 signaling in the cardiomyocyte.

## 1. Introduction

Pathological cardiac remodeling is regulated by a network of signaling pathways that can induce hypertrophic growth and affect survival, resulting in the dysfunction of the myocyte contractility, metabolism, and gene expression [1,2]. Despite the extensive characterization of many of these pathways, how this network is differentially modulated in response to pressure- and volume-overload diseases remains unclear [3,4]. In this context, the Rho family of small GTPases, which regulates actin organization, cell movement, polarity, growth and survival, and gene expression [5], serves important roles in regulating the cardiac myocyte structure and function and the control of cardiac hypertrophy [6,7]. For instance, increased Rac1 activity is associated with myocyte hypertrophy both in vitro and in vivo, and Rac1 knockout mice are protected against Angiotensin II (Ang II)-induced pathological hypertrophy [8]. This role in hypertrophy is associated with Rac1′s unique function among the Rho family members to increase the NADPH oxidase activity and thus increase oxidative stress, and as result, exacerbate cardiac hypertrophy [9]. In contrast, Cdc42 cardiac myocyte-specific knockout mice (CKO) have a normal baseline phenotype, but they present exaggerated pathological remodeling and heart failure in response to pressure overload, enhanced hypertrophy following chronic Ang II and phenylephrine (PE) infusion, and a propensity towards sudden death during exercise [10]. The protective role of Cdc42 in pressure-overload disease appears to result from the activation of c-Jun N-terminal kinase (JNK), which can oppose hypertrophy by the phosphorylation of the calcineurin-dependent nuclear factor of activated T cells (NFAT) transcription factor [11,12]. Accordingly, the transgenic overexpression of MAP kinase kinase 7 (MKK7) to restore the activity of JNK could rescue the effects of Cdc42 CKO following pressure overload [10].

The signaling activity of Rho-family GTPases is mediated by >100 potential effectors, and it is regulated by both the cycling of GTP/GDP binding and localization to membranes by C-terminal isoprenylation [6]. The Dock (Dedicator of cytokinesis) family of proteins consists of 11 guanine exchange factors (GEFs) for the Rho family of GTPases. Dock10, a ubiquitously expressed member of subgroup D of the Dock family, has GEF activity for both Cdc42 and Rac1 in vitro [13,14]. Dock10 is most highly expressed in leukocytes and appears to be important for the development and function of B cells [15,16,17,18,19,20]. In neurons, Dock10 is required for dendritic spine morphogenesis by a mechanism involving Cdc42, but not Rac1 [14]. In addition, Dock10 siRNA inhibits the amoeboid invasion of melanoma cells via the downregulation of Cdc42, N-WASP, and PAK2 signaling, and simultaneously increases the Rac1 activity [21]. In vitro, Dock10 overexpression can activate both Cdc42 and Rac1 [13,22]. Both Cdc42 and Rac1, as opposed to RhoA, activate JNK and p38 MAPK [23,24,25]. Thus, it is likely that, through its GEF activity for Cdc42 and Rac1, Dock10 can modulate JNK and p38 signaling. Indeed, the activation of the Rac1/JNK signaling axis downstream of Dock10 was recently shown to modulate epithelial-to-mesenchymal transition [26].

Dock10 function has not been studied in the heart, where it might serve as an important regulator of hypertrophic signal transduction. Several studies have described mice with global Dock10 KO as viable and fertile, with no overt signs of pathology [18,27,28]. Given a recent report that indicates the increased longevity of Dock10 KO [27], it is unlikely that Dock10 is required for physiologic cardiac function. However, because Dock10 regulates Cdc42 and Rac1, we aimed to determine the roles played by this protein in the heart under both baseline conditions and in the presence of pathological stress. Here, we show that Dock10 indeed regulates the p38 and JNK activity of heart cells and is vital for cardiac function, which opposes the induction of pathological cardiac remodeling. According to an analysis of Dock10 CKO mice, Dock10’s role in opposing the induction of pathological cardiac remodeling is related to cardiac cell types other than the myocyte.

## 2. Results

### 2.1. Dock10 Ablation Differentially Affects p38 and JNK Signaling in NMCMs and NMCFs

We evaluated neonatal mouse cardiac myocytes (NMCMs) and fibroblasts (NMCFs) from *Dock10^flox/flox^* mice for Dock10 expression following transduction with adenovirus expressing Cre-recombinase (Figure 1). We confirmed the genomic DNA recombination by PCR using specific primers for the floxed and null alleles (Figure 1A,B). According to the Western blots, we confirmed a marked reduction in the Dock10 protein expression in both types of cells six days post-transduction (Figure 1C–F).

We evaluated the requirement of Dock10 expression for p38 and JNK signaling in both NMCMs and NMCFs, as both signaling pathways are known downstream effectors of Rho GTPases, including cdc42 and Rac1 [10,23,24,25,29,30]. We stimulated the cells with the α-adrenergic agonist phenylephrine (PE) for 20 min before cell lysis. As expected, the exposure of the control NMCMs to PE led to the increased phosphorylation of both p38 and JNK (1.94 ± 0.28 fold and 1.57 ± 0.20 fold, respectively; *p* < 0.01 for both). In contrast, in the Dock10 KO NMCMs, the baseline level of p-p38 was somewhat elevated (Figure 2A), and the p-p38 response to PE was markedly blunted (Figure 2A,C). Although we also noted a tendency for a reduction in the response of p-JNK to PE, it did not reach significance (Figure 2B,D). In NMCFs, a somewhat similar effect of PE was noted in the control cells, with the PE increasing the phosphorylation of both p38 and JNK (1.43 ± 0.09 fold and 2.01 ± 0.23 fold, respectively; *p* < 0.001 for both). However, Dock10 ablation specifically inhibited the p-JNK signaling in response to PE without a notable effect on p-p38 (Figure 3). Thus, it appears that Dock10 differentially affects p38 and JNK in these different types of cells (see discussion).

### 2.2. Mice with Global Dock10 KO Demonstrate a Subtle Phenotype of Cardiac Dysfunction

Although we had difficulty detecting the Dock10 protein by Western blot in whole-heart extracts, the level of Dock10 mRNA was reduced by 70% in the Dock10 KO mice, presumably due to nonsense-mediated mRNA decay. Dock10 KO was not associated with the altered expression of the D-family members Dock9 and Dock11 (Figure 4A). In previous reports, the authors describe the Dock10 KO mice as having no overt phenotype [18]. Accordingly, a gravimetric analysis of the Dock10 KO mice showed no difference from the controls in body weight and heart-to-body weight ratio (Figure 4B). However, according to echocardiography, the Dock10 KO mice had mildly elevated diastolic and systolic LV diameters (Figure 4C) and a small but significant decrease in fractional shortening (Figure 4D), implying mild systolic dysfunction.

Moreover, the expressions of fetal genes associated with cardiac stress were elevated in the Dock10 KO mice relative to the controls (Figure 4E), including those for natriuretic factors. To corroborate the defect in the cardiac contractility, we acutely isolated cardiomyocytes from control and KO mice, and we evaluated them by IonOptix calcium and contractility measurements. Under constant pacing, the Dock10 KO cardiomyocytes demonstrated an increased resting sarcomere length, decreased systolic shortening, reduced fractional shortening (FS), and a slower rate of cardiomyocyte relaxation (Figure 5A). In addition, Dock10 KO cardiomyocytes had increased diastolic Ca^2+^ levels, as well as a slower rate of both Ca^2+^ rise and decline during the transient (Figure 5B). Thus, while the Dock10 KO did not induce cardiac hypertrophy, according to functional studies and fetal gene analysis, the ablation of this Rho GEF induced a mild dilated cardiomyopathy in unstressed mice.

### 2.3. α-Adrenergic-Induced MAPK Signaling in the Heart Depends on Dock10 Expression

To further delineate the role of Dock10 in cardiac signal transduction, we investigated whether the MAPK signaling alterations observed in vitro in neonatal cardiac cells (Figure 2 and Figure 3) could be detected in vivo in adult cardiac tissue. We acutely treated Dock10 KO and WT littermate mice with PE or saline, and we analyzed the phosphorylation of p38 and JNK by Western blot. As expected, the PE induced prominent p38 and JNK phosphorylation in WT mice (2.89 ± 0.37 fold and 7.54 ± 0.67 fold, respectively; *p* < 0.001 for both). The responses of both signaling pathways to PE stimulation were markedly attenuated by the Dock10 KO (Figure 6). Thus, in addition to the mild functional phenotype described above, these mice also demonstrate an apparent biochemical phenotype that is consistent with the abnormal activation of Rho family kinases (see discussion).

### 2.4. Global Dock10 KO Exacerbates Angiotensin II-Induced Pathological Cardiac Remodeling

In light of the above results, we hypothesized that the cardiac phenotype of global Dock10 KO mice might be further impaired under chronic neurohormonal-stress conditions. To test this hypothesis, we exposed three-month-old mice for two weeks to a chronic infusion of Ang II or saline control. As expected, the Ang II induced cardiac hypertrophy and mild systolic dysfunction in the WT mice. Notably, Ang II induced a more prominent phenotype in Dock10 KO mice with further increased hypertrophy and systolic dysfunction (Figure 7, Appendix A). Accordingly, the Dock10 KO mice exposed to Ang II demonstrated a markedly higher level of interstitial fibrosis (Figure 8A,B), increased mRNA expression levels of collagen genes (Figure 8C), and the enlargement of the cardiomyocyte cross-section area (Figure 8D).

### 2.5. Cardiomyocyte-Specific KO of Dock10 Does Not Exacerbate Pathological Cardiac Phenotype

Based on the above findings that suggest a role for Dock10 in the regulation of pathological cardiac remodeling, we considered whether this function might be attributed to Dock10 in the cardiomyocyte. To test whether Dock10 has a cardiomyocyte autonomous function in pathological remodeling, we compared the cardiac mass and cardiac systolic function of Dock10 CKO (Tg(Myh6-cre;*Dock10^flox/flox^*) mice with two relevant littermate controls: Tg(Myh6-cre) and *Dock10^flox/flox^* mice. According to the baseline measures, there was no difference between the Dock10 CKO and controls (not shown). Moreover, following two weeks of chronic infusion of Ang II, the Dock10 CKO mice did not differ in any gravimetric or echocardiographic parameters from the two control groups (Appendix A), which implies that the observed pathological phenotype in the global Dock10 KO mouse was not due to the loss of Dock10 in the cardiomyocytes.

## 3. Discussion

Dock proteins are GEFs that activate small GTPases of the Rho protein family, regulating various crucial cellular functions in multiple cell types and organs. Dock-family members participate in a wide range of physiological processes [31], including vascular development [32]. However, to the best of our knowledge, our study is the first to investigate the role of Dock10 in the myocardium. The central findings of this study are as follows: (a) Dock10 ablation differentially modulated the activation of p38 and JNK pathways in NMCMs and NMCFs, and inhibited the responses of both signaling pathways in adult heart ventricular tissue; (b) Global Dock10 KO mice demonstrate a baseline phenotype of mild cardiac dysfunction, which is markedly exacerbated by long-term exposure to Ang II; (c) In contrast to the Dock10 global KO mice, the Dock10 CKO mice did not exhibit an obvious cardiac phenotype.

Previously, researchers demonstrated the antihypertrophic role of the Rho GTPase cdc42 [10], and its downstream effectors, JNK [11] and p38 [33], in cardiomyocytes. In the current study, the loss of Dock10 markedly affected the p38 and JNK signaling in the heart following Gq stimulation by PE. Interestingly, Dock10 was differentially responsible for activating p38 in neonatal cardiomyocytes, and JNK in cardiofibroblasts. In addition, according to an analysis of the Dock10 KO mouse, the activation of these signaling pathways was Dock10-dependent in the adult heart, as shown previously in other cells and tissues in vitro [13,14].

Besides affecting JNK and p38 signaling, the Dock10 KO was associated with mildly reduced systolic function, increased fetal gene expression, and abnormal diastolic Ca^2+^ handling, which are consistent with the decreased SERCA2a Ca^2+^ reuptake activity, as might be found in early stages of heart failure [34,35]. Moreover, and consistent with the baseline phenotype, the Dock10 KO increased the hypertrophic response, decreased the systolic function, and increased cardiac fibrosis induced by chronic Ang II infusion. However, the absence of a similar phenotype in the Dock10 CKO mouse suggests that the role of Dock10 in maintaining normal cardiac function probably lies in cells other than cardiomyocytes. Indeed, while alterations in the Cdc42 and Rac1 signaling of cardiomyocytes affect the development of cardiac hypertrophy and heart failure, these Rho GTPases can have important roles in other cell types of crucial importance in the heart, including cardiofibroblasts [36,37] and endothelial cells [38,39]. In addition, the well-documented function of Dock10 in immune cells, including macrophages [16,17,19], may also be considered as a possible mechanism that leads to the observed phenotype in the global Dock10 KO mice [40]. Further studies are required to properly differentiate between these possibilities. The hypertrophic response and remodeling observed in the global Dock10 mice possibly occurred because of the accumulated impact of the loss of function of this protein in several different cardiac cell types combined.

In conclusion, in this study, we define a critical role for Dock10 in the maintenance of the normal cardiac structure and function and the inhibition of pathological cardiac remodeling. These data shed additional light on an important anti-hypertrophic pathway in the heart, providing information on the roles of GEFs and Rho GTPases in the heart. A further understanding of such pathways in cardiomyocytes and other cardiac-related cells may help with the development of future novel therapeutic strategies for heart failure.

## 4. Limitations

The main limitation of our study is that we did not test the direct activation of cdc42 and Rac1, the direct targets of Dock10 function as a GEF. However, the established ability of cdc42 and Rac1 to activate JNK and p38 [23,24,25], as well as the previously documented ability of Dock10 to activate the Rac1/JNK signaling axis [26], strongly support the notion that the diminished activity of these MAPK signaling pathways in our study was a result of reduced cdc42 and/or Rac1 activation. A further mechanistic understanding of the differential effects of Dock10 that we identified in myocytes and fibroblasts will require direct investigation in future studies. In addition, our study is limited by the description of the effects of Dock10 on MAPK signaling only following PE stimulation. Indeed, it would be of interest to study the acute effects of other pharmacological agents that stimulate Gq signaling, such as Ang II in NMCMs, NMCFs, and in vivo. However, the acute effects of PE and Ang II on these pathways are generally similar in vitro [41]. Thus, in the present study, we chose to focus specifically on PE due to its stronger, more sustained, and well-documented acute activation of these pathways [41,42].

## 5. Methods

### 5.1. Animal Care

We housed the mice under standardized conditions throughout the study, according to the home office guidelines: 12:12 light:dark cycles at 20–24 °C and 30–70% relative humidity. We free-fed the animals autoclaved rodent chow, and they had free access to reverse-osmosis-filtered water. We monitored the mice on a daily basis for signs of stress or inappropriate weight loss, according to guidance from the Ben-Gurion University veterinary services (assured by the Office of Laboratory Animal Welfare, USA (OWLA) #A5060-01, and fully accredited by the Association for Assessment and Accreditation of Laboratory Animal Care International (AAALAC)). At the end of all the experiments, we euthanized the animals under deep anesthesia.

### 5.2. Global Dock10 KO Model

We acquired mice with a “knockout first allele” (reporter-tagged insertion with conditional potential) [43] of Dock10 (EMMA mouse repository, EM:04723) after back-crossing 10 times with C57BL/6J mice. While these Dock10 KO mice have previously been described as lacking Dock10 in the lung, spleen, and thymus [18], this KO is based on splicing acceptors and might not be fully efficient in all tissues. We could not detect Dock10 expression by the Western blot of adult cardiac tissue. Thus, to further ensure full Dock10 deletion in the cardiomyocytes, we crossed these mice with the B6N.FVB(B6)-Tg(Myh6-cre)2182Mds/J (Jackson laboratory #018972) mouse that expresses Cre-recombinase under the control of the α-myosin heavy-chain promoter (Tg(Myh6-cre) mice). The Cre-mediated recombination of the knockout first allele results in the deletion of exon 4 and frameshift mutation, which prevent the translation of the protein after amino acid 111. We back-crossed Tg(Myh6-cre) mice at least 5 times with C57BL/6J mice. All the global KO experiments compared male Tg(Myh6-cre); Dock10 knockout first allele^+/+^ mice with male Tg(Myh6-cre) littermates.

### 5.3. Dock10 CKO Model

For the in vitro experiments and for the creation of cardiac-specific Dock10 CKO mice, we crossed mice containing the Dock10 knockout first allele (EM:04723) one time with a mouse expressing FLP recombinase (JAX 009086) to generate *Dock10^flox/flox^* mice. We created cardiac-specific Dock10 KO mice by crossing the *Dock10^flox/flox^* mice with Tg(Myh6-cre) mice. In vitro, we induced Dock10 recombination in primary cultures using an adenovirus expressing Cre-recombinase, as detailed below.

### 5.4. Genotyping

We performed polymerase chain reactions (PCRs) using primers purchased from Sigma-Aldrich (St. Louis, MO, USA); for the wild-type allele (740 bp) and floxed allele (892 bp): Dock10-5′arm (gctacatggtataggtaggtg) and Dock10-3′arm (taatcctcagacatcctccag). For the Dock10 knockout first allele (631 bp): Dock10-5′arm and LAR3 (caacgggttcttctgttagtcc); for the null allele (944 bp): Dock10-5′arm and Dock10-3′as (ctggaagagagtgtaagtag).

### 5.5. Osmotic Mini Pumps

Three-month-old mice were implanted with Osmotic mini pumps (Model 2004, ALZET, Cupertino, CA, USA) to provide Ang II (2 mg/kg/day in saline) or the saline control for 14 days. After 2 weeks, we sacrificed the animals under deep pentobarbital anesthesia. We separated the ventricles in cold PBS, and we weighted and divided them into samples snap-frozen in liquid nitrogen or fixated for histological sectioning.

### 5.6. Echocardiography

We performed the echocardiography under light isoflurane anesthesia and strict temperature control using a Vevo 3100 ultrasound (FUJIFILM VisualSonics, Toronto, ON, Canada), as previously described by our group [44]. All the measurements were performed in a blinded manner by a skilled technician. The maximum duration of the echocardiography procedure was 15 min. We obtained 2D images of the left ventricle in parasternal long- and short-axis views. We took long- and short-axis M-mode images at the mid papillary muscle level, with cursor penetration at the papillary muscle tip. We evaluated the LV end-systolic diameter (LVID) and LV end-diastolic diameter (LVIDd) from the long-axis M-mode trace. We performed calculations of the LV fractional shortening (FS) (%) according to (LVIDd-LVIDs)/LVIDd × 100. We calculated the LV ejection fraction (LVEF) using planimetry as follows: EF = 100 × (LVIDd^3^ − LVIDs^3^)/LVIDd^3^.

### 5.7. Acute α-Adrenergic Stimulation

We intraperitoneally injected two-month-old mice with 10 mg/kg PE or saline. At thirty minutes postinjection, we sacrificed the mice, and we dissected their hearts in cold PBS and snap-froze them in liquid nitrogen for later biochemical analysis.

### 5.8. Calcium and Contractility Measurements from Isolated Cardiomyocytes

We recorded the sarcomere length and intracellular calcium [Ca^2+^]_i_ signals using the HyperSwitch dual-excitation and dual-emission photometry system (IonOptix, Westwood, MA, USA). We isolated the cardiomyocytes using conventional enzymatic dissociation. We incubated the cells for 10 min in HEPES-buffered Tyrode’s solution (TB) (in mmol/L: NaCl: 140; KCl: 5.4; MgCl_2_: 1; CaCl_2_: 1; HEPES: 10; glucose: 10 (pH 7.4 at 37 °C)) that contained 10 µM Indo-1AM (Molecular Probes, Eugene, OR, USA) and Pluronic F-127 (Life Technologies, Carlsbad, CA, USA) at a dilution of 1:1, followed by a 10 min wash with TB to remove excess dye. We transferred the cells to a perfusion chamber mounted on the stage of an inverted microscope, and we perfused them with TB for 4–5 min to stabilize them. We evaluated the mechanical contraction by sarcomere-length measurements, and we obtained [Ca^2+^]_i_ signals by the Indo-1 fluorescence ratio (R = F405/F480; indicative of [Ca^2+^]_i_). We simultaneously recorded both signals during the stimulation at 0.5 Hz using a field stimulator (MyoPacer; IonOptix, Westwood, MA, USA) through two platinum electrodes placed on the sides of the perfusion chamber. We analyzed the data using an IonWizard data-acquisition system (IonOptix, Westwood, MA, USA). We calculated the parameters from an average of 15 to 20 successive transients, approximately 50–80 cells, from 3 mice per cohort.

### 5.9. Dock10 Ablation in Primary Cultures of Neonatal Mouse-Heart Cells

We produced the primary cultures of neonatal mouse cardiomyocytes (NMCMs) by the enzymatic dissociation of 0–3-day-old *Dock10^flox/flox^* mouse hearts. We collected the hearts into ice-cold M-199 media (Biological Industries, Inc., Beit Haemek, Israel), and we gently washed them and cut them into 1–2 mm^3^ sections in calcium and magnesium-free PBS (Biological Industries, Inc.). We subjected the cardiac-tissue pieces to 5–7 rounds of enzymatic digestion using digestion buffer containing 0.3 mg/mL pancreatin (Sigma-Aldrich, MO, USA) and 45 units/mL of collagenase type 2 (Worthington, NJ, USA), dissolved in ADS buffer consisting of: 116 mM NaCl; 1 mM NaH_2_PO_4_; 5.5 mM glucose; 5.39 KCl; 1 mM MgSO_4_ × 7H_2_O; 20 mM HEPES. We collected cells from each round by centrifugation at 600× *g* for 4 min at 4 °C, and resuspension in M-199 media supplemented with 0.5% fetal bovine serum, 100 units/mL penicillin/streptomycin, and 2 mmol/L L-glutamine (Biological Industries, Inc., Israel). We combined all the fractions, centrifuged them for 5 min at 4 °C, and we replaced the media with plating media consisting of M-199 supplemented with 5% fetal bovine serum, 100 units/mL penicillin/streptomycin, and 2 mmol/L L-glutamine. To reduce nonmyocyte cells, we preplated the cell suspension for 75 min on an uncoated cell-culture dish, and we then plated the suspension in 12-well gelatin-coated dishes for 24 h before replacing with a new plating media. After 48 h in culture, we transduced the NMCMs with 250 MOI of purified (Vivapure, Sartorius, Gottingen, Germany) adenovirus expressing either GFP (control) or Cre-recombinase (a gift from Prof. Itzhat Kehat) to induce Dock10 KO. At six days postinfection, we exposed the cells to 50 µM of PE or the equivalent volume of saline for 20 min before final lysis.

We resuspended the cells attached to the preplating dish (mostly cardiofibroblasts, e.g., NMCFs) in fresh DMEM supplemented with 10% fetal bovine serum, 100 units/mL penicillin/streptomycin, and 2 mmol/L L-glutamine, and the dish was incubated for 72–96 h until we achieved 90% confluence. We then separated the NMCFs by trypsin and plated them in 12-well dishes. At 24 h postplating, we transfected the cells with Cre-recombinase or GFP expressing adenovirus (MOI 400). At six days postinfection, we exposed the cells to 50 µM PE for 20 min, as described above for the NMCMs.

### 5.10. Histological Analysis

We fixed transverse sections from the mid-base of the hearts in 4% paraformaldehyde for 24 h, embedded them in paraffin, and cut them into 5 µm-thick slices. We deparaffinized the slices in xylene, rehydrated them in a descending alcohol sequence, and imported them into distilled water. We conducted the Masson trichrome staining according to the manufacturer’s instructions (04-010802, Bio Optica, Milano, Italy). We imaged the stained sections with a panoramic scanner (panoramic MIDI II, 3DHISTH, Budapest, Hungary), and we automatically analyzed them by customized software (Quant center version 2.0 software, 3DHISTH, Budapest, Hungary). The collagen volume fraction (CVF) reported the cardiac fibrosis, and we calculated it with the equation of total collagen area/total field area (including both perivascular and interstitial collagen). We analyzed 5 images, covering the entire base section in each of 4 mice per experimental group. We measured the surface area across centered cardiomyocytes using ImageJ software version 1.52a, National Institutes of Health, Bethesda, Maryland, USA. From each experimental group, we analyzed 184–229 cells from at least 3 different mice. We performed all of the histological analysis in a blinded manner, and we marked the measured cells to avoid repeated measurements of the same cell.

### 5.11. Western Blotting

We homogenized cells or tissues in RIPA buffer that contained protease and phosphatase inhibitors and prepared them for Western blotting, as previously described [45]. We incubated the membranes overnight with primary antibodies: rabbit anti-phospho-p38 (Catalog # 4511S), rabbit anti-p38 (Catalog # 9212), mouse anti-phospho-JNK (Catalog # 9255S), and rabbit anti-JNK (Catalog # 9252S), all from Cell Signaling Technology (Danvers, MA, USA). Dr. Antonio Parrado provided the rabbit anti-Dock10 antibody [13]. For the loading control, we used a mouse anti-vinculin antibody (va131; Sigma). We washed the membranes three times after primary antibody incubation in TBST, and we then incubated them with horseradish peroxidase-conjugated secondary antibodies, which were species-specific (anti-mouse (7076) and anti-rabbit (7074), both from Cell Signaling Technology), and we then washed them three times. We detected the bands by a Chemiluminescence substrate (WESRAR ɳC XLS100,0500 or WESTAR SUPERNOVA XLS3,0100; Bio-lab, Jerusalem, Israel), and we acquired the images using an ImageQuant LAS4000 imager (GE Healthcare, Uppsala, Sweden). We performed the densitometry using ImageJ (version 1.52a) software.

### 5.12. Gene-Expression Analysis by Quantitative PCR

We homogenized the hearts with TRI-reagent (Zymo), and we isolated the RNA using Z Direct-zol RNA MiniPrep (R2050, Zymo Research, Irvine, CA, USA). We measured the concentration and purity with a Nano Drop 1000 spectrophotometer (Thermo Fisher Scientific, Waltham MA, USA). We performed the synthesis of cDNA using random hexamers and Taqman reverse-transcription reagents, according to the manufacturer’s protocol (95047 Quanta Bioscience, Beverly Hills, CA, USA). We performed the gene-expression assays with reverse transcriptase-quantitative polymerase chain reaction (RT-qPCR) and PerfeCTa SYBR Green FastMix (Quanta BioSciences, Beverly Hills, CA, USA) using a Real Time PCR System Instrument—7300 (Applied Biosystems, Waltham, MA, USA). We ran the reactions in the program of preincubation: 50 °C for 3 min and 95 °C for 3 min, followed by 40 cycles of 95 °C for 10 s and 60 °C for 45 s, and a final stage of dissociation. We calculated the relative gene expression by the efficiency 2^−ΔΔCT^ method, with the expressions of the genes of interest (Appendix A) normalized to the GAPDH housekeeping gene.

### 5.13. Statistical Analysis

We express the values as mean ± SEM. We performed the statistical analyses using Prism 8.0 (GraphPad Software, San Diego, CA, USA). We performed the comparisons between the two groups using an unpaired Student’s t-test. In cases in which n was lower than 6, we performed a Mann–Whitney test instead. We analyzed experiments with multiple groups using one-way ANOVA and Tukey post hoc testing. If n was lower than 6, then we performed a Kruskal–Wallis with Dunn’s multiple comparisons post-test instead. We mention the specific tests that we used in the legend of each figure. The criterion for significance was set at *p* < 0.05. The *p*-values are displayed graphically as follows: * *p* ≤ 0.05, ** *p* ≤ 0.01, and *** *p* ≤ 0.001.

## Figures and Tables

**Figure 1 ijms-23-09616-f001:**
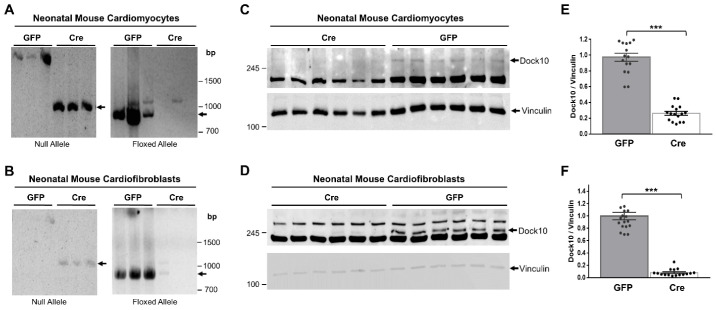
Confirmation of Dock10 ablation in primary murine cardiac cells. We transduced neonatal mouse cardiomyocytes (NMCMs) and neonatal cardiofibroblasts (NMCFs) derived from mice with floxed Dock10 alleles with an adenovirus expressing Cre-recombinase or the GFP control. We prepared DNA samples from the cells 5 days post virus infection. We established two PCR reactions; primers for the null allele, which is produced upon the deletion of exon 4 of Dock10, and primers for the floxed allele. (**A**,**B**) Null- and floxed-allele PCR reactions in NMCMs and NMCFs, respectively. (**C**,**D**) Example of Western blot analyses confirming Dock10 protein knockout post-adenoviral transduction in NMCMs and NMCFs, respectively. We used Vinculin as a loading control. (**E**,**F**) Densitometric analysis of Dock10 expression following adenoviral transduction in NMCMs and NMCFs, respectively. *n* = 16–18 in each condition, obtained from 3 independent experiments. We performed the statistical analyses using the Mann–Whitney test. *** *p* ≤ 0.001.

**Figure 2 ijms-23-09616-f002:**
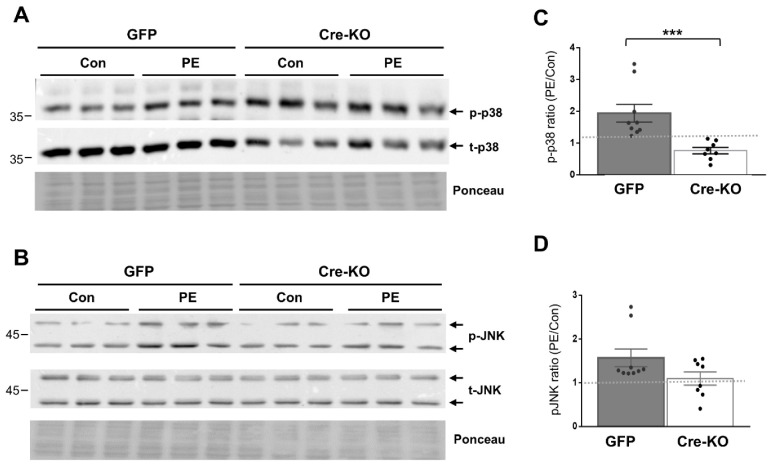
Dock10 ablation in NMCMs inhibits p38 signaling following acute exposure to PE. We evaluated NMCMs transduced with adenovirus expressing Cre-recombinase or GFP following incubation with the α-adrenergic agonist PE (50 μM, 20 min). (**A**,**B**) Western blots for total and phosphorylated p38 and JNK. Ponceau staining of the same blots was performed as internal loading controls with representative images shown here. (**C**,**D**) Quantitation of the PE-induced phosphorylation of p38 and JNK relative to saline control treatment (control). Note the marked inhibition of PE-induced p38 phosphorylation in the Dock10 KO cells. We performed pairwise comparisons using the Mann–Whitney test. *** *p* ≤ 0.001.

**Figure 3 ijms-23-09616-f003:**
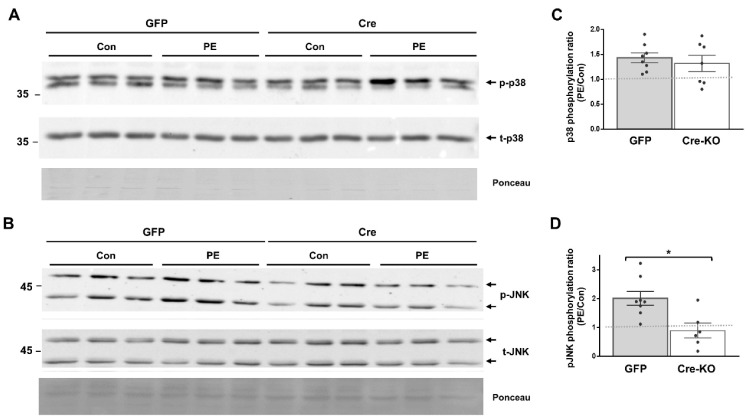
Dock10 ablation in NMCF inhibits JNK signaling following acute exposure to PE. We evaluated NMCFs transduced with adenovirus expressing Cre-recombinase or GFP following incubation with the α-adrenergic agonist PE (50 μM, 20 min). (**A**,**B**) Western blots for total and phosphorylated p38 and JNK. Ponceau staining of the same blots was performed as internal loading controls with representative images shown here. (**C**,**D**) Quantitation of the PE-induced phosphorylation of p38 and JNK relative to saline control treatment (control). Note the marked inhibition of PE-induced JNK phosphorylation in the Dock10 KO cells without a notable effect on p38 signaling. We performed the statistical analysis using the Mann–Whitney test. * *p* ≤ 0.05.

**Figure 4 ijms-23-09616-f004:**
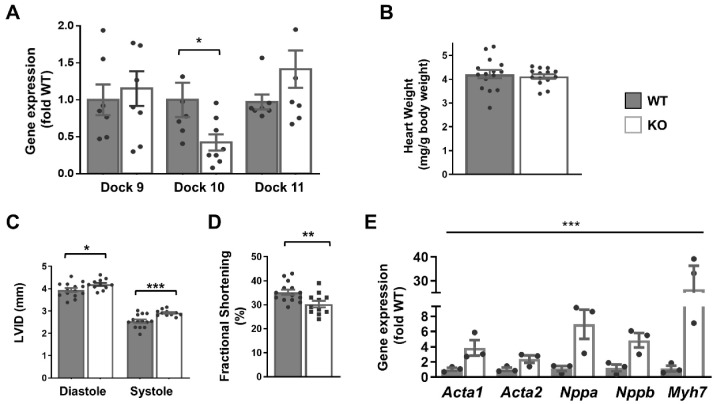
Global Dock10 KO mice exhibit mild cardiac dysfunction. (**A**) q-PCR analyses for Dock 9, Dock10, and Dock11 in the ventricular tissue of Global Dock10 KO (KO) mice and littermate controls. Results were normalized to GAPDH and are presented as fold relative to WT gene expression. For used primers, see Appendix A. We performed the statistical analysis using the Mann–Whitney test. (**B**–**D**) Twelve-week-old mice (*n* = 11–15) were studied by gravimetric analysis and echocardiography. HW/BW: heart-weight–body-weight ratio (mg/g); LVIDd and LVIDs: left ventricular interior diameter in diastole and systole, respectively (mm); FS: fractional shortening (%). We performed the statistical analysis using an unpaired t-test. (**E**) RT-qPCR analyses as in (**A**) for expression of the fetal genes *Acta1*, *Acta2*, *Nppa*, *Nppb*, and *Myh7*. *** *p* ≤ 0.001, ** *p* ≤ 0.01, * *p* ≤ 0.05.

**Figure 5 ijms-23-09616-f005:**
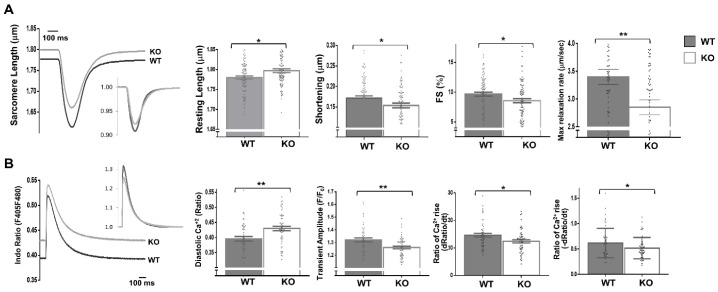
Dock10 KO cardiomyocytes exhibit impaired contractile function and Ca^2+^ handling. We analyzed dissociated ventricular myocytes using an IonOptix HyperSwitch system. (**A**) Representative traces for sarcomere length during field stimulation (0.5 Hz); inset shows traces normalized to baseline sarcomere length. Parameters shown are resting sarcomere length, shortening (difference between resting and contracted length), fractional shortening (shortening/resting length), and maximal relaxation rate (*n* = 87, Global Dock10 KO, *n* = 95, W.T. control). (**B**) Representative traces for cytosolic Ca^2+^ as assayed using the ratio-metric dye Indo-1/AM, and emission measured at 405 nm (Ca^2+^ bound) and 480 nm (Ca^2+^ free), shown as relative terms (a ratio of F405/F480). Inset shows traces normalized for diastolic (resting) Ca^2+^. Parameters shown are diastolic (resting) Ca^2+^, transient amplitude, the rate of Ca^2+^ rise at the beginning of the transient and the rate of Ca^2+^ decline at the end of the transient (*n* = 53, Global Dock10 KO, *n* = 50, W.T. control). The analyzed cells were obtained from 3 mice per cohort. We performed the statistical analysis using the Mann–Whitney test. ** *p* ≤ 0.01, * *p* ≤ 0.05.

**Figure 6 ijms-23-09616-f006:**
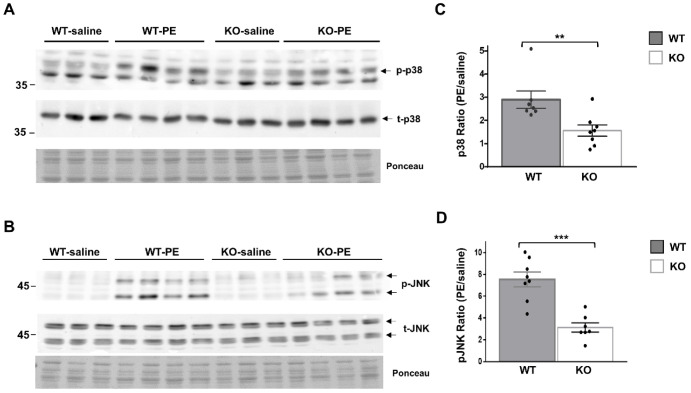
Global Dock10 KO mice exhibit attenuated cardiac JNK and p38 signals following acute exposure to PE. (**A**,**B**) Western blots for total and phosphorylated p38 and JNK from 2-month-old mice-heart lysates after exposure to PE (10 mg/kg) or saline (control). Ponceau staining of the same blots was performed as internal loading controls with representative images shown here. (**C**,**D**) Quantitation of the PE-induced phosphorylation of p38 and JNK relative control treatment. Note the marked inhibition of the PE-induced phosphorylation of both p38 and JNK in the global Dock10 KO hearts. We performed the statistical analysis using the Mann–Whitney test. *** *p* ≤ 0.001, ** *p* ≤ 0.01.

**Figure 7 ijms-23-09616-f007:**
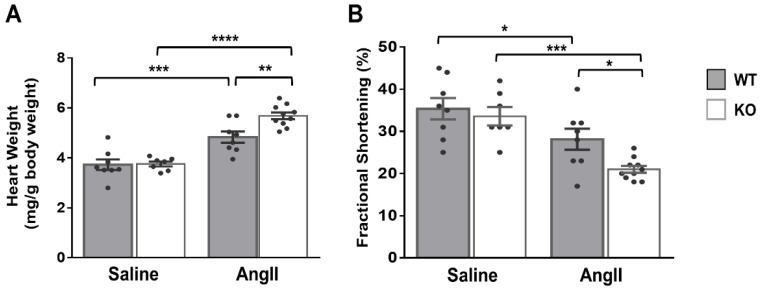
Dock10 KO mice exhibit exacerbated pathological hypertrophy and cardiac dysfunction following chronic exposure to Ang II. We subjected Dock10 KO and WT mice to chronic Ang II infusion at 2 mg/kg/day for 2 weeks. (**A**) Heart weight normalized by body weight (mg/g). (**B**) Fractional shortening by M-mode echocardiography. We performed the statistical analysis by one-way ANOVA with Tukey post hoc multiple comparisons. **** *p* ≤ 0.0001, *** *p* ≤ 0.001, ** *p* ≤ 0.01, * *p* ≤ 0.05.

**Figure 8 ijms-23-09616-f008:**
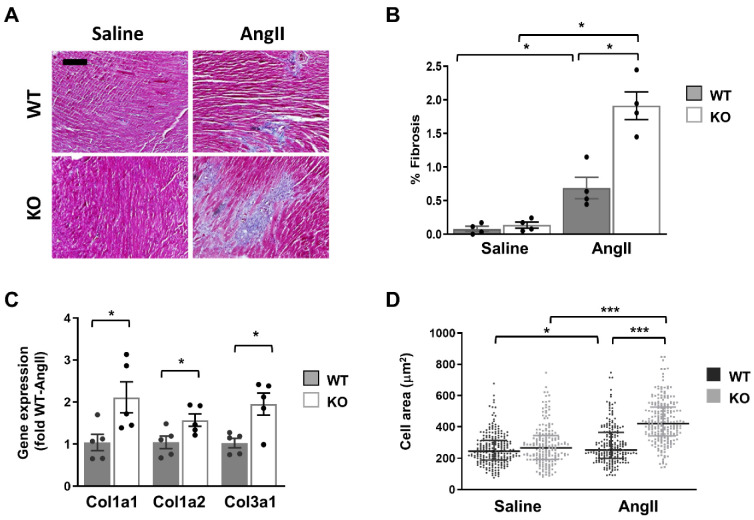
Dock10 KO mice exhibit exacerbated characteristics of pathological cardiac hypertrophy following chronic exposure to Ang II. (**A**) Masson trichrome-stained, paraffin-embedded sections for Ang II-infused and saline-infused, WT and Dock10 KO hearts. Black bar = 100 µm. (**B**) Quantitative analysis of the percentage of fibrosis in the ventricular tissue. We performed the analysis on 5 images, covering the entire LV base for each of 4 mice per experimental group. (**C**) RT-qPCR for collagen gene expression. Data are normalized to GAPDH and presented as fold relative to Ang II-treated WT mice (*n* = 5 per cohort). We performed the statistical analysis using the Mann–Whitney test. (**D**) Analysis of cardiomyocyte cell area. Note the markedly increased cross-section area of myocytes from Ang II-treated global Dock10 KO mice. We analyzed a total of 184–229 cells for each condition from *n* = 3–4 mice per cohort. We performed the statistical analysis by ordinary one-way ANOVA with Tukey post hoc multiple comparisons. *** *p* ≤ 0.001, * *p* ≤ 0.05.

## Data Availability

The data will be made available upon request.

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
