# Peer review of "Dock10 Regulates Cardiac Function under Neurohormonal Stress"

_ijms, 2022, doi:10.3390/ijms23179616_

Round 1
Reviewer 1 Report
The authors investigated the role of pathological cardiac remodeling of Dock10, a guanine nucleotide exchange factor for Cdc42 and Rac1 that regulates MAPK signaling. In vitro, Dock10 ablation differently inhibited α-adrenergic stimulation of p38 and JNK in NMCM and NMCF, respectively. In vivo, surprisingly, Dock10 KO exaggerated the pathological response to AngII infusion, but Dock10 CKO did not. Although Dock10 can regulate MAPK signaling in both myocardial and myocardial fibroblasts, inhibition of pathological cardiac remodeling is clearly not due to Dock10 signaling in myocardial cells. These results are interesting in the context of cardiac remodeling. However, the authors need to address a few methodological issues, and provide explanation for several important observations.
1.The authors showed that Dock10 KO mice exacerbated cardiac fibrosis after chronic exposure to AngII.  The authors need to examine the effect of the Dock10 ablation model for NMCF on p38 and JNK signaling after AngII stimulation, as well as on proliferation and secretion of any cytokines or growth factors.
2.  The authors should investigate the effect of Dock10 ablation on Rac and cdc42 activation in cardiomyocytes and fibroblasts after AngII and PE stimulation.
3.The authors should show the internal controls in the Western blots in Figures 2, 3 and 6.
4. The authors indicate that Dock10 ablation inhibited α-adrenergic stimulation of p38 and JNK in NMCM. What is the effect of PE administration in Cardiomyocyte-specific KO of Dock10?
5. Fig1 is the same as Fig2. The authors need to revise.
Reviewer 2 Report
Segal et al. investigated the roles of Dock10, a guanine-nucleotide exchange factor for Rho GTPases that regulate mitogen-activated protein kinase (MAPK) signaling, in the heart. They demonstrated that global Dock10 knockout mice exhibited mild cardiac dysfunction, which was exacerbated by pathological conditions induced by angiotensin II treatment. Additional studies using cultured cardiomyocytes/cardiofibroblasts and cardiomyocyte-specific Dock10 knockout mice raised the possibility that the roles of Dock10 diverse depending on the cell types in the heart.
This study is interesting and provides a new insight into the roles of Dock10 in the cardiac function and diseases. However, this reviewer considers that this manuscript has important concerns before acceptation for publication in International Journal of molecular Sciences.
1) There are many careless mistakes, both major and minor, throughout the manuscript. For instance, Figure 1 and 2 are totally the same. There are various writings for one word; DOCK10, Dock10, and Dock10 (italic). Abbreviation was explained in the second appearance of the word, not in the first. There are more mistakes. The manuscript should be submitted after more careful reading and editing by all the authors.
2) Introduction should include the outline of the cascade which shows the relationship among Dock10, Rho, Rac, Cdc42, and MAPK. And/or an additional figure, which summarizes the hypothesized roles of Dock10 and schemes of the related pathways in the heart, should be added.
Round 2
Reviewer 1 Report
The authors have improved the manuscript with revisions and I have no further comments to make.
Author Response
We were glad to see that the revisions are satisfactory and we thank the reviewer again for the important input.
Reviewer 2 Report
The manuscript has been well revised according to the reviewers. As described in my original comment 2, an additional figure, which summarizes the hypothesized roles of Dock10 and schemes of the related pathways in the heart, should be added. Plus, there are still many careless mistakes. The text should be edited by the professional editing service.
Author Response
Comment #1: In the revised MS we added a graphical abstract which summarizes the hypothesized roles of Dock10 and schemes of the related pathways in the heart.
Comment #2: The MS was edited by the professional editing service.